# Age-Stage, Two-Sex Life Tables of the Predatory Mite *Cheyletus Malaccensis* Oudemans at Different Temperatures

**DOI:** 10.3390/insects11030181

**Published:** 2020-03-12

**Authors:** Weiwei Sun, Miao Cui, Liyuan Xia, Qing Yu, Yang Cao, Yi Wu

**Affiliations:** Academy of National Food and Strategic Reserves Administration, National Engineering Laboratory of Grain Storage and Logistics, Henan Collaborative Innovation Center of Grain Crops, Jiangsu Collaborative Innovation Center for Modern Grain Circulation and Safety, No. 11 Baiwanzhuang Street, Beijing, 100037, China; sww@chinagrain.org (W.S.); cm@chinagrain.org (M.C.); xly@chinagrain.org (L.X.); yuq@chinagrain.org (Q.Y.); cy@chinagrain.org (Y.C.)

**Keywords:** *Cheyletus malaccensis* Oudemans, biological control, age-stage two-sex life table

## Abstract

*Cheyletus malaccensis* Oudemans is a predatory mite inhabiting grain depots in China. The relationship between temperature and the population growth rate of *C. malaccensis* is useful for predicting its population dynamics. Age-stage, two-sex life tables of the predator, *C. malaccensis*, reared on *Acarus siro* were constructed under laboratory conditions at 22, 24, 28, 30, and 32 °C, 75% relative humidity, and a 0:24 h (L:D) photoperiod. Increasing temperature shortened the development time of the immature stages. The complete generation time of *C. malaccensis* ranged from 11.10 d to 27.50 d. Life table parameters showed that 28 °C was the optimum temperature for the growth and development of *C. malaccensis*; populations could increase rapidly at this temperature. The highest net reproductive rate (*R*_0_ = 290.25) and highest fecundity (544.52) occurred at 28 °C. Temperature significantly affected the intrinsic rate of increase (r), fecundity, and finite rate of increase (λ). The values of age-specific fecundity (high to low) were 28 °C > 24 °C > 30 °C > 32 °C > 22 °C, while the values of age-stage-specific fecundity had the same trend.

## 1. Introduction

*Cheyletus malaccensis* Oudemans is a predator mite species in China. It preys on acarid grain mites and small arthropods, such as the eggs and first-instar larvae of stored grain pests [1,2,3]. *C. malaccensis* populations can be self-sustaining for limited time periods by cannibalism [4,5]. *C. malaccensis* is the dominant predaceous mite species in grain depots [6,7,8,9,10] and provides biological control of common pests in stored grain [11]. *C. malaccensis* development includes five stages: egg, larvae, protonymph, deutonymph, and adult. The deutonymph is absent in males. Virgin females always produce males, whereas fertilized females produce both male and female progeny [12,13].

Temperature is an important component of predator–prey interactions, as it influences pest and natural enemy population dynamics such as developmental time [14], life span [15], reproductive rate [16], and control efficiency [17,18,19]. Temperature affects the survival and development of mites [20], and seasonal temperature variations also affect predator–prey interactions [12]. For *C. malaccensis*, the female life span (egg to adult) is longer (20–23 d) than the male life span (15–17 d) at 25 °C [21]. Palyvos and Emmanouel [12] studied the life history of *C. malaccensis* at six constant temperatures: 17.5, 20, 25, 30, 32.5, and 35 °C. The life span was 53.0 d at 17.5 °C, and 15.4 d at 35 °C. Toldi et al. [22] found that fecundity was highest at 25 °C with the value of 415.62 ± 24.78 eggs/female, and lowest at 20 °C. Thus, temperature has a significant influence on the development and reproduction of *C. malaccensis* [14,23].

Life tables can be used for predicting the population levels of pests and predators as well as the efficiency of biological control [24]. Insect-related life table technology is a technical method used to study population dynamics [16,25,26]. However, the traditional life table of *C. malaccensis* only addresses the females, while ignoring the males, life stage differentiation, and variable developmental rates. This is a limited practical application of the life table. The age-stage, two-sex life table is a superior alternative since it considers males and different age individuals in the population [27,28] and systematically studies the growth, development, survival rate, reproduction, and pesticide susceptibility of males and females. To quantify the effect of temperature on the development of *C. malaccensis*, life stages of *C. malaccensis* were held at constant temperatures and the life history raw data were analyzed based on an age-stage, two-sex life table. The age-stage, two-sex life table technology has also been used to predict the population growth and predation rate of other natural enemies [29].

To understand the relationship between temperature and the population growth rate of *C. malaccensis*, we constructed age-stage, two-sex life tables of *C. malaccensis* fed on *Acarus siro* Linnaeus at different temperatures. Specifically, we studied (a) the life history of *C. malaccensis* at 22, 24, 28, 30, and 32 °C and 75% RH, and (b) the influence of temperature on the development and reproduction of *C. malaccensis*. The results provided basic information for biological control programs that use *C. malaccensis*.

## 2. Materials and Methods

### 2.1. Insect Rearing

*C. malaccensis* was initially collected from Haikou, Hainan Province, China and reared at the Institute of Grain Storage & Logistics Academy of National Food and Strategic Reserves Administration at 28 °C, 75% RH, and a photoperiod of 0:24 h (L:D). The mites were identified based on morphological characteristics [1].

*Acarus siro* was provided by the Crop Research Institute, Prague, Czech Republic and was reared on whole wheat flour, under constant conditions (28 °C, 75% RH, and 0:24 (L:D)).

### 2.2. Life Table Study of C. Malaccensis

Fifty female *C. malaccensis* adults were randomly selected and the mites were reared in plastic micro rearing cells (20 × 20 × 2 mm) at 28 °C and 75% RH, with *A*. *siro* as prey food. In the center of each block, a conical shaped hole was drilled. A piece of black filter paper (20 × 20 mm) was attached to the lower surface of the cell and a suitable glass cover slip was placed on its upper surface (Figure 1).

After 24 h, 50 eggs were collected and designated as the F1 generation for further study. To determine the optimum development temperature, experiments were conducted at five temperatures (22, 24, 28, 30, and 32 °C), with 75% RH. The F1 generation eggs from each adult were randomly selected as a cohort to construct the corresponding life table.

Eggs were individually placed inside blocks and subjected to different temperatures (22, 24, 28, 30, and 32 °C) at 75% RH. *A. siro* were used as food and 15–25 *A. siro* were added daily for each *C. malaccensis*. Each block was checked daily for eggs. The egg incubation period, development times of immature mites, survival rates of larvae and adults, and fecundity of females (number of eggs laid) were recorded daily.

### 2.3. Life Table Analysis

The raw data for *C. malaccensis* individuals were analyzed on the basis of the age-stage, two-sex life table theory [27,28] using a TWOSEX-MSChart-2018.11.01 [30] (http://140.120.197.173/Ecology/prod02.htm).

Because of the absence of male deutonymphs, all nymphal stages were referred to as “nymph”. The four stages considered in the growth and development of *C. malaccensis* were egg, larva, nymph, and adult. The age-stage-specific survival rate ($S_{xj}$) (probability that a newly laid egg will survive to age x and stage j), the age-stage-specific fecundity ($f_{xj}$) (number of hatched eggs produced by female adult at age x, and j is the life stage number (j = 4), the age-specific survival rate ($l_{x}$) (probability that a newly laid egg will survive to age x), the age-specific fecundity curve ($m_{x}$) (the average fecundity of the individuals), and the age-stage life expectancy ($e_{xj}$) (expected time that an individual of age x and stage j is expected to live) were calculated as follows [31,32]:(1)$$l_{x}=\overset{\beta }{\underset{j=1}{\sum }}S_{xj}$$
(2)$$m_{x}=\frac{\sum_{j=1}^{\beta }S_{xj}f_{xj}}{\sum_{j=1}^{\beta }S_{xj}}$$
(3)$$e_{xj}=\overset{\infty }{\underset{i=x}{\sum }}\overset{\beta }{\underset{y=j}{\sum }}S_{iy}^{\prime }$$

The net reproductive rate ($R_{0}$), the mean generation time (T), the intrinsic rate of increase ($r_{m}$), and the finite rate of increase (λ) were also calculated as follows [27]:(4)$$R_{0}=\sum l_{x}m_{x}$$
(5)$$T=\frac{\ln R_{0}}{r_{m}}$$
(6)$$\overset{\infty }{\underset{x=0}{\sum }}e^{-r_{m}\left(x+1\right)}l_{x}m_{x}=1$$
(7)$$\lambda =e^{r_{m}}$$

### 2.4. Statistical Analysis

The raw life history data for *C. malaccensis* obtained for each of the temperature regimes were entered separately into a Microsoft Excel 2016 data sheet. One-way ANOVA was used to study the effect of temperature on the development time of immature stages and the longevity of *C. malaccensis*. The means, standard errors, and variances of the population parameters were estimated using the bootstrap technique [33,34,35] (10,000 samples), which is contained in the TWOSEX-MSChart program. Differences among different temperatures were compared using the Tukey–Kramer procedure. Excel 2016 was used to create $S_{xj}$, $f_{x4}$, $l_{x}$, $m_{x}$, $l_{x}m_{x}$ and $e_{xy}$ curves.

## 3. Results

### 3.1. Life History Study

*C. malaccensis* females and males completed development from egg to adult emergence at constant 22 to 32 °C temperatures (Table 1). The egg incubation period of *C. malaccensis* ranged from 1.90 to 5.25 d for females, and from 1.80 to 5.43 d for males. The shortest developmental time for the egg stage was at 32 °C and the longest was at 22 °C in both females and males. The egg incubation duration and the larva duration of females were significantly longer at 22 °C than at the other temperatures with similar results in males (*p* < 0.05). The nymph period was not significantly different (*p* < 0.05) at the five different temperatures, with the shortest developmental time occurring at 32 °C and the longest at 22 °C in both females and males. The life history, both in females and males, exhibited a significant difference at 22 °C compared to the other temperatures (*p* < 0.05); it ranged from 11.10 to 27.50 d (females) and 8.80 to 22.71 d (males). The development time of male adults was shorter than females within the experimental temperature range and the development duration decreased with increased temperature.

Within the temperature range studied, the development time of females first increased and then decreased (Figure 2). Development time was >50 d at 22–28 °C, and longest (66 d) at 28 °C. The development time of males decreased gradually in the range of 22–32 °C, with a maximum of 46.71 d at 22 °C. Based on the total duration, males developed more quickly than females at all temperatures.

These results showed that increasing the temperature generally shortened the development time of *C. malaccensis*. Considering the fecundity and adult period, 24–28 °C was the ideal temperature range for reproduction and biological control use of *C. malaccensis*.

### 3.2. Population Parameters

Table 2 shows the population parameters of *C. malaccensis*, based on the two-sex life table theory and analysis technology, at 22, 24, 28, 30, and 32 °C with 75% RH. The mean generation time and life expectancy of *C. malaccensis* decreased gradually with an increased temperature. The lowest values at 32 °C were 12.49 d (mean generation time) and 22.15 d (life expectancy). The maximal $R_{0}$ (290.25) occurred at 28 °C. The $r_{m}$ of *C. malaccensis* increased with increasing temperature. The minimum was 0.12 at 22 °C, and the maximum was 1.4 at 32 °C. λ showed the same tendency and ranged from 1.12 to 1.40.

### 3.3. Life Table Analysis

Because of the absence of deutonymphs in males, the nymphal stages were referred to as “nymph”. Figure 3 shows age-stage-specific survival rates ($S_{xj}$), which indicate the rate of individuals surviving to age x and stage j. The $S_{xj}$ curves varied greatly at different temperatures and overlaps were observed in the $S_{xj}$ curves, which demonstrated the variable developmental rates among individuals. The eggs of *C. malaccensis* hatched at all of the temperatures, and the incubation time decreased significantly with an increased temperature. The mean generation time of *C. malaccensis* shortened with an increasing temperature, from 100 d at 22 °C to 31 d at 32 °C. Mite survival was highest at 28 °C and lowest at 32 °C. The results showed that there were overlapping generations in the growth and development of *C. malaccensis*. Excessively high temperature had adverse effects on the growth and development of *C. malaccensis*.

Figure 4 summarizes the age-specific survival rate ($l_{x})$, the age-specific fecundity ($m_{x})$, the age-specific reproductive value ($l_{x}m_{x}),\;$and the age-stage-specific fecundity ($f_{x4})$ of *C. malaccensis* at different temperatures. The age-specific survival rate ($l_{x})$ simplified the survival of different development periods and did not consider differences among individuals. At 22 °C, the age-specific survival rate ($l_{x})\;$of *C. malaccensis* showed a trend from 0 to 56 d, and slowly decreased to 80%. After 56 d, the survival rate decreased rapidly from 80% to 0% (Figure 4A). At 24 °C, the survival rate of *C. malaccensis* decreased slowly to 88% from 0 to 43 days and then rapidly decreased to 0% at the age of 95 d (Figure 4B). At 28 °C, the age-specific survival rate decreased from 100% at 21 d to 80% at 31 d. It then rapidly decreased to 40% at 75 d and 0% at 88 d (Figure 4C). The age-specific survival rate curves of 30 °C and 32 °C showed the same trend of steady decline during the early development stages and then rapid decline near the end of development (Figure 4D,E).

At the experimental temperatures, the $m_{x}$ curve had similar trends with the $f_{x4}$ curve. At 22 °C, the reproductive peaks of $f_{x4}\;$and $m_{x}$ occurred at the age of 30 d (Figure 4A), whereas the reproductive peaks occurred at 63 d at 24 °C (Figure 4B). At 28 °C, the peak of $f_{x4}$ occurred at 21 d, while the $m_{x}$ curve reached a reproductive peak at 50 d (Figure 4C). The $f_{x4}$ curves reached reproductive peaks early in the oviposition periods (16 d under 30 °C and 11 d under 32 °C) (Figure 4D,E). The highest peak values of the $f_{x4}$ and $m_{x}$ curves were at 28 °C, whereas the lowest values were at 22 °C, with peak values (high to low) at 28 °C > 24 °C > 30 °C > 32 °C > 22 °C. These results showed that the fecundity of *C. malaccensis* was highest and the population growth was most rapid at 28 °C.

Figure 5 shows the age-stage specific life expectancy (the time that an individual of *C. malaccensis* of age *x* and stage *j* is expected to live) ($e_{xj}$) of *C. malaccensis* at different temperatures. The ($e_{xj}$) of *C. malaccensis* gradually decreased to 0 as age increased. At 22, 24, 28, and 32 °C, the $e_{xj}$ of female adults of *C. malaccensis* was higher than male adults during the whole development stage. At 30 °C, the $e_{xj}$ of males was lower than females except for the ages of 18–26 d, but higher than females in other development periods. The $e_{xj}$ curve decreased synchronously in both males and females at 30 °C. The $e_{xj}$ at 28 °C was slightly higher than the other temperatures. The $e_{xj}$ values of the initial reproducing *C. malaccensis* fed on *A. siro* were 65.79, 59.50, 56.93, 27.54, and 22.15 at 22, 24, 28, 30, and 32 °C, respectively, which was also the average life expectancy of the individuals. The life expectancies of *C. malaccensis* at 30 and 32 °C were about 50% of those at 22, 24, and 28 °C. These results show that temperatures from 22 to 28 °C were best for the growth and development of *C. malaccensis*.

## 4. Discussion

### 4.1. Temperature

Temperature is the most important environmental factor determining the development and reproduction of arthropods [12,36], and temperature affects the immature time, longevity, fecundity, and survival rates of arthropods [37,38]. 

The immature period development time decreased with increased temperature, which is consistent with other reports describing the effects of temperature on the growth and development of *C. malaccensis* [13,15,21,22]. The immature period development was slower compared to the findings of Palyvos and Emmanouel [15] at 25 °C and 30 °C, using *Tyrophagus putrescentiae* as prey. The life history was shorter than that reported by Saleh [21] when the growth temperature was 25 °C and *Aleuroglyphus ovatus* was the prey. These differences may be due to the different prey used and may be related to prey quantity and nutrition quality provided by the prey. In addition, the efficiency of detecting and accessing can also cause differences in the results. Thus, it would be useful to study the effects of alternative prey for *C. malaccensis*. The predatory potential of cheyletidae mites has been reported, and mass rearing of cheyletidae mites in the laboratory has been described [18,39]. Many studies on the predator *Cheyletus eruditus* (Schrank) have been published [39,40,41]. Compared to *C. eruditus*, *C. malaccensis* is better adapted to higher temperatures, and therefore may have greater potential for biological control in warmer grain storage environments. As temperature increases, growth increases up to an optimum point after which higher temperatures begin to have negative effects on development; this observation is consistent with this study. When the temperature exceeded 28 °C, the adult lifespan decreased rapidly in both males and females. High temperature also had negative effects on the development of *C. malaccensis*. The environments of different ecological regions vary greatly in China. In actual application, it is best to make changes in predator numbers according to the different seasons and the different geographical area. To establish the predator population before a pest outbreak, and to ensure it will be sufficient for successful biological control, predatory mites need to be released prior to the development of the pest problem. In addition, it is advisable to make a safety evaluation of different predatory mites and to establish an optimum ratio of predators to their prey.

The adult is the most predatory stage of *C. malaccensis* [14]. We found that an optimal feeding temperature can shorten the development time of *C. malaccensis*. The development time was >50 d during a temperature range of 22 to 28 °C, with the longest development time being 66 d at 28 °C. Considering fecundity and duration of the adult period, 24–28 °C is an ideal temperature range for reproduction and growth of *C. malaccensis*. 

### 4.2. Life Table

The intrinsic rate of increase ($r_{m}$) includes the population survival rate, duration of development, and fecundity. It is an important life table parameter and reflects the population growth capability at different temperatures [42]. Life table parameters can predict the future development trend of the population. Based on the two-sex life table theory and analysis technology, age-stage two-sex life tables of *C. malaccensis* fed on *A*. *siro* were constructed at temperatures ranging from 22 to 32 °C at 75% RH. The biological parameters, including growth, development, and fecundity, of *C. malaccensis* at different temperatures were compared and analyzed using the TWOSEX-MSChart.

The population growth ability of *C. malaccensis* was highest at temperatures ranging from 24 to 28 °C. All individuals were included in the analysis, including both females and males. The $r_{m}$ of *C. malaccensis* increased with increasing temperature. This tendency is consistent with previous studies [14,15,18]. The $r_{m}$ and *λ* of *C. malaccensis* were highest at 32 °C, while $R_{0}$ and the single female fecundity of *C. malaccensis* were highest at 28 °C. However, the fecundity of *C. malaccensis* fed on *T. putrescentiae* was highest at 25 °C and 30 °C for virgin females, and the highest fecundity for previously mated females was at 30 °C [15]. These differences may be related to different biotypes in different regions of the world. Filipponi [43] reported that some mite species may have different reproductive patterns in different geographic regions. Other possible causes of variation include the ambient temperatures and the prey species used to feed the mites. Before the 1980s, *C. eruditus* was reported to be the dominant species in China [44]. However, *C. malaccensis* is now reported to be the dominant predaceous mite species with the highest potential for biocontrol in grain depots in China [6,7,8,9,10]. In contrast, *C. malaccensis* was reported as the mite species having the lowest potential for biocontrol in the Czech Republic [5]. This discrepancy may be due to differences in climate, geographical environments, and population replacement.

Our data indicate that *C. malaccensis* can develop and reproduce well at temperatures ranging from 22 to 32 °C. However, in life table studies, we cannot measure the growth potential of a population based on single parameters. It is thus necessary to conduct a comprehensive analysis of the growth and development parameters of the entire population. To predict the times and the number of releases for biological control, it is useful to understand the development rate, age differentiation, reproductive rate, and survival rate of *C. malaccensis*. Life tables can help us understand the comprehensive effects of various factors on the population growth of *C. malaccensis*. Traditional life tables have typically focused on the female population and lacked the contribution of males to population growth. This has limited their practical application. The stage differentiation and stage overlaps in development can be accurately described by using the age-stage, two-sex life table, whereas the traditional life table is incapable of accomplishing this. Under normal conditions, females make a higher contribution to the population and have a higher life expectancy and survival rates at all stages compared to males. Because of this, most current life table studies have focused on females [45,46].

Male adults also contribute to predation and there were many differences between females and males. We found that the survival rate of male adults was higher than that of female adults at specific temperatures and specific ages. This indicated that male adults may have a higher survival rate than female adult mites at certain temperatures. More studies on males are needed to confirm these findings. This will be helpful for the establishment of populations in biological control programs and for high-temperature and low-temperature regimes in the artificial propagation of *C. malaccensis*. It is necessary to determine the long-term effects in consideration of the age-stage, two-sex life table. The comprehensive evaluation of a predatory natural enemy requires consideration of the basic parameters affecting its growth and development, reproduction, and population dynamics. More attention needs to be paid to predator–prey interactions under natural conditions. Increased knowledge of *C. malaccensis* biology will increase its utility as a biological control agent.

## 5. Conclusions

Age-stage, two-sex life tables of *C. malaccensis*, reared on *Acarus siro* were constructed at 22, 24, 28, 30, and 32 °C, 75% relative humidity. Increasing temperature shortened the development time. The complete generation time of *C. malaccensis* ranged from 11.10 d to 27.50 d. The optimum temperature for the growth and development of *C. malaccensis* was 28 °C. Populations could increase rapidly, occurred the highest net reproductive rate ($R_{0}$ = 290.25) and highest fecundity (544.52) at this temperature. The values of age-specific fecundity (high to low) were 28 °C > 24 °C > 30 °C > 32 °C > 22 °C. The result is useful for predicting its population dynamics, and guiding artificial breeding and delivery *C. malaccensis* to control the stored-product pests. 

## Figures and Tables

**Figure 1 insects-11-00181-f001:**
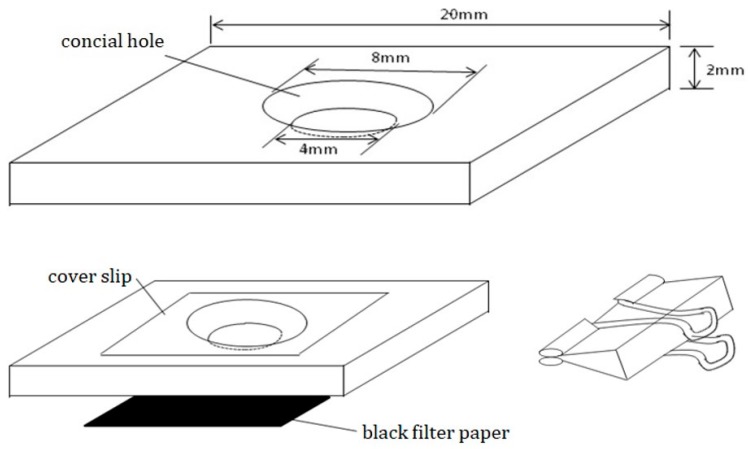
Feeding container and schematic diagram for use.

**Figure 2 insects-11-00181-f002:**
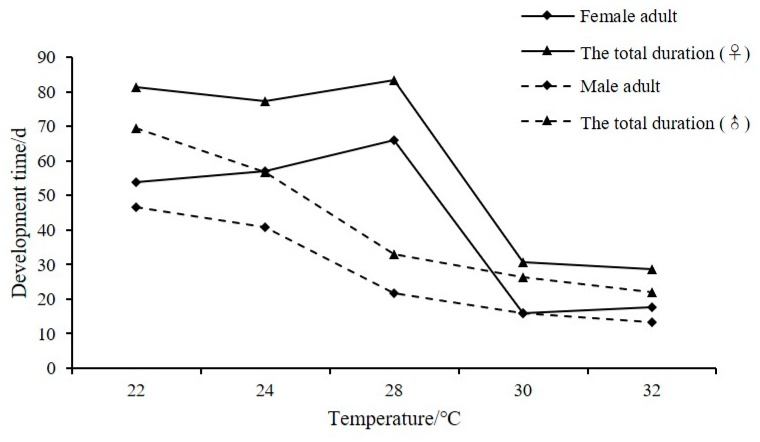
Relationship between temperature and the developmental time of *C. malaccensis.*

**Figure 3 insects-11-00181-f003:**
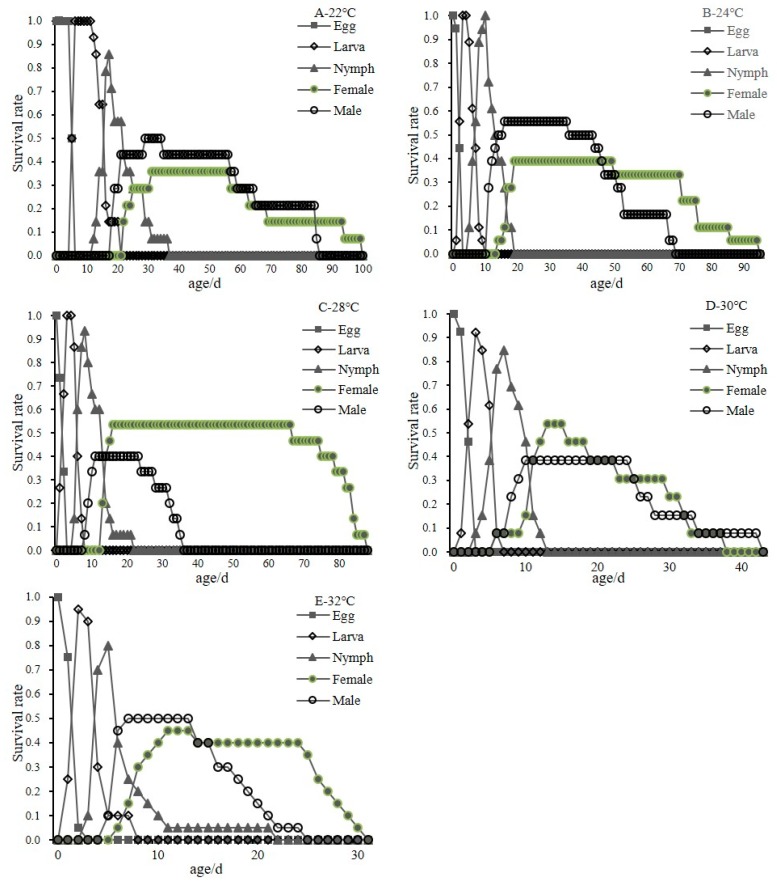
Age-stage specific survival rate ($S_{xj}$) of *C. malaccensis* at different temperatures (A-E are 22, 24, 28, 30 and 32 °C, respectively).

**Figure 4 insects-11-00181-f004:**
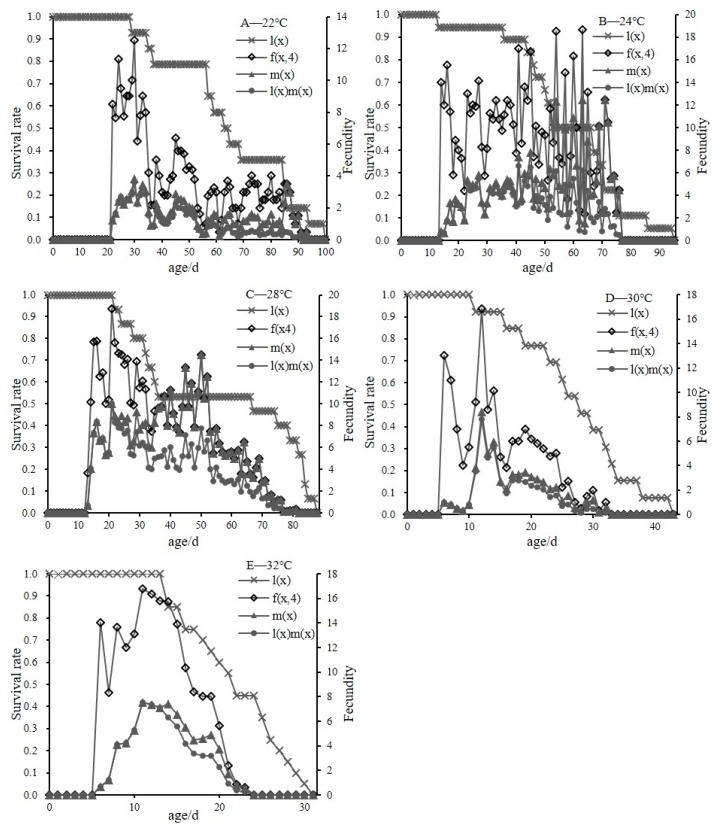
Age-specific survival rate ($l_{x}$) fecundity ($m_{x}$), maternity ($l_{x}m_{x}$), and age-stage specific fecundity ($f_{x4}$) of *C. malaccensis* at different temperatures (A-E are 22, 24, 28, 30 and 32 °C, respectively).

**Figure 5 insects-11-00181-f005:**
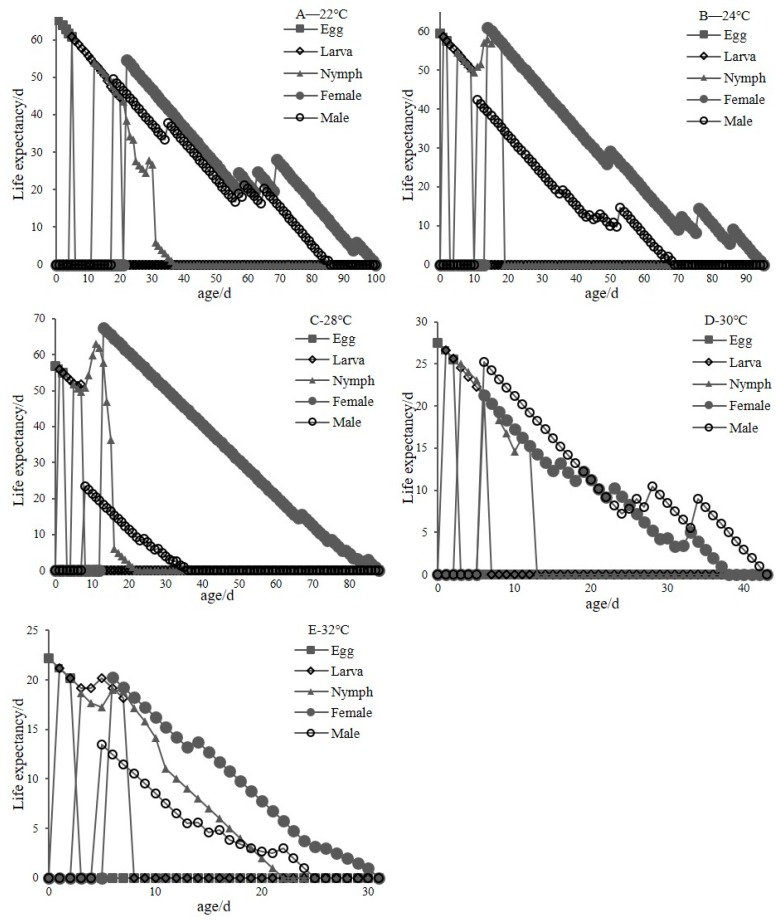
Age-stage specific life expectancy ($e_{xj}$) of *C. malaccensis* at different temperatures (A-E are 22, 24, 28, 30 and 32 °C, respectively).

**Table 1 insects-11-00181-t001:** Development time of *C. malaccensis* reared at different temperatures under laboratory conditions (M ± SD) (d).

Temperature/°C	Egg		Larva		Protonymph		Deutonymph		Life history		Adult	
	Female	Male	Female	Male	Female	Male	Female	Male	Female	Male	Female	Male
22	5.25 ± 0.25a	5.43 ± 0.20a	9.75 ± 0.48a	10.00 ± 1.02a	4.25 ± 0.95a	5.29 ± 0.61a	5.25 ± 0.95a	—	27.50 ± 2.18a	22.71 ± 1.46a	54.00 ± 9.75a	46.71 ± 6.70a
24	3.00 ± 0.00b	2.40 ± 0.15b	4.30 ± 0.33b	6.00 ± 0.68b	4.70 ± 0.33a	5.10 ± 0.31a	5.30 ± 0.33a	—	20.30 ± 0.33b	15.90 ± 0.73b	57.00 ± 21.59a	40.80 ± 5.69ab
28	2.30 ± 0.29b	1.80 ± 0.40b	4.30 ± 0.18b	4.20 ± 0.17bc	3.30 ± 0.18a	3.30 ± 0.21a	4.60 ± 0.30a	—	17.50 ± 0.48b	11.30 ± 0.33c	66.00 ± 2.96a	21.80 ± 1.85c
30	2.40 ± 0.24b	2.00 ± 0.58b	3.40 ± 0.51b	3.00 ± 0.58c	2.80 ± 0.37a	3.33 ± 0.33a	3.00 ± 0.55a	—	14.60 ± 0.40b	10.33 ± 1.20bc	16.00 ± 3.97b	16.00 ± 5.29bc
32	1.90 ± 0.20b	1.80 ± 0.15b	3.60 ± 0.34b	3.60 ± 0.18c	2.70 ± 0.24a	2.40 ± 0.18a	3.00 ± 0.29a	—	11.10 ± 0.39b	8.80 ± 0.15c	17.60 ± 1.58b	13.20 ± 1.05c

**Note**: Data in the table are represented as mean ± SE. The means followed by different letters in the same columns are significantly different at the 0.05 level based on one-way ANOVA and Tukey’s HSD multiple range test.

**Table 2 insects-11-00181-t002:** Population parameters of *C. malaccensis* at different temperatures.

Temperature/°C	T	$R_{0}$	$r_{m}$	λ	Fecundity	Life Expectancy
22	34.80 ± 2.83cd	77.42 ± 32.61ab	0.12 ± 0.02d	1.13 ± 0.02d	216.20 ± 54.03a	65.80 ± 5.75c
24	30.30 ± 1.97c	204.75 ± 60.33b	0.18 ± 0.02c	1.19 ± 0.02c	526.15 ± 6.85cd	59.51 ± 4.48c
28	24.51 ± 0.66b	290.25 ± 70.58b	0.23 ± 0.15b	1.26 ± 0.19b	544.52 ± 13.47c	56.91 ± 6.68c
30	14.37 ± 1.31a	50.28 ± 15.79a	0.27 ± 0.03ab	1.31 ± 0.04ab	93.41 ± 17.66b	27.55 ± 2.38b
32	12.49 ± 0.65a	67.03 ± 18.02a	0.33 ± 0.03a	1.40 ± 0.04a	148.96 ± 16.10a	22.15 ± 1.22a

**Note**: Data in the table are represented as mean ± SE. The means followed by different letters in the same columns are significantly different at the 0.05 level based on one-way ANOVA and Tukey’s HSD multiple range test. $R_{0}$ net reproductive rate, T mean generation time, $r_{m}$ intrinsic rate of increase, λ the finite rate of increase.

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
