# Peer review of "Age-Stage, Two-Sex Life Tables of the Predatory Mite Cheyletus Malaccensis Oudemans at Different Temperatures"

_insects, 2020, doi:10.3390/insects11030181_

Round 1
Reviewer 1 Report
This ms deals with the relationships between temperature and population growth of a predatory mite Cheyletus malaccensis with relative humidity of 75% and a 0:24 photoperiod. This mites were fed with Acarus siro.
I think that this ms include some data of publication quality.
Althogh others papers from China has been published on predatory mites, the main contribution of this work is consider males in the application of the life table. This information could be basic for biological control programs.
After having made a detailed and critical reading of the text, we have come to the following conclusions: firstly, the methods are laborious and appropriate to the ends to be achieved. The work is well structured and the results sustain the conclusions presented. The references employed have been updated in the various sections. Therefore, only two suggestions for facilitating the reading of the text have been made: firstly, the merging of tables 1 and 2 into just one table and secondly, submitting the graphs as Supplementary Material.
These recommendations are commented in the text.

Author Response
Thank you for your suggestions. All of the suggestions were valuable for improving our manuscript (MS). We have provided a detailed response to each comment in the attachment.

Reviewer 2 Report
please see comments on manuscript.

Author Response

(The authors gave the same response as above.)

Round 2
Reviewer 2 Report
Much improved.
there parts of the study that are not completely clear. I found the study sufficient for publication.